# Centrality dependence of multistrange baryon production in high-energy heavy-ion collisions

G.H. Arakelyan[†1], C. Merino[2⋆] and Yu.M. Shabelski[3]

**1** A.Alikhanyan National Scientific Laboratory (Yerevan Physics Institute)
Yerevan, Armenia
**2** Dpto. de Física de Partículas, Facultade de Física
Instit. Galego de Física de Altas Enerxías (IGFAE)
Universidade de Santiago de Compostela, Galiza, Spain
**3** Petersburg Nuclear Physics Institute, NCR Kurchatov Institute
Gatchina, St.Petersburg, Russia
* carlos.merino@usc.gal

January 23, 2023

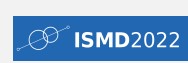 *51st International Symposium on Multiparticle Dynamics (ISMD2022)*
## Abstract

We consider the experimental data on the production of strange $\Lambda$'s, and multistrange baryons ($\Xi$, $\Omega$), and antibaryons, on nuclear targets, at the energy region from SPS up to LHC, in the framework of the Quark-Gluon String Model. One remarkable result of this analysis is the significant dependence on the centrality of the collision of the experimental $\overline{\Xi}^+/\overline{\Lambda}$ and $\overline{\Omega}^+/\overline{\Lambda}$ ratios in heavy-ion collisions, at SPS energies.

## Contents

---

[†]Deceased

# 1 Introduction

We compare [1] the results obtained in the Quark-Gluon String Model (QGSM) formalism, with the corresponding experimental data on yields of $\Lambda$, $\Xi$, and $\Omega$ baryons, and the corresponding antibaryons, in nucleus-nucleus collisions, for a wide energy region going from SPS up to LHC. We also consider the ratios of multistrange to strange antihyperon production in nucleus-nucleus collisions with different centralities, in the same energy range.

The QGSM [2] is based on the Dual Topological Unitarization, Regge phenomenology, and nonperturbative notions of QCD. In QGSM, high energy interactions are considered as proceeding via the exchange of one or several Pomerons. The cut of at least some of those Pomerons determines the inelastic scattering amplitude of the particle production processes. In the case of interaction with a nuclear target, the Multiple Scattering Theory (Gribov-Glauber Theory) is used. At very high energies, the contribution of enhanced Reggeon diagrams leads to the suppression of the inclusive density of secondaries into the central (midrapidity) region. The QGSM provides a succesful description of multiparticle production in hadron-hadron, hadron-nucleus, and nucleus-nucleus collisions, for a wide energy region.

The production of multistrange hyperons, $\Xi^-$ (dss), and $\Omega^-$ (sss), has special interest in high energy particle and nuclear physics. The production of each additional strange quark featuring in the secondary baryons, i.e., the production rate of secondary $B(qqs)$ over secondary $B(qqq)$, then of $B(qss)$ over $B(qqs)$, and, finally, of $B(sss)$ over $B(qss)$, is affected by one universal strangeness suppression factor, $\lambda_s$:

$$\lambda_s = \frac{B(qqs)}{B(qqq)} = \frac{B(qss)}{B(qqs)} = \frac{B(sss)}{B(qss)} \,, \tag{1}$$

together with some simple quark combinatorics.

Let us define:

$$R(\overline{\Xi}^+/\overline{\Lambda}) = \frac{dn}{dy}(A+B \rightarrow \overline{\Xi}^+ + X)/\frac{dn}{dy}(A+B \rightarrow \overline{\Lambda} + X) \,, \tag{2}$$

$$R(\overline{\Omega}^+/\overline{\Lambda}) = \frac{dn}{dy}(A+B \rightarrow \overline{\Omega}^+ + X)/\frac{dn}{dy}(A+B \rightarrow \overline{\Lambda} + X) \,. \tag{3}$$

The ratios in eqs. (2) and (3) can be calculated in the QGSM as a function of the strangeness suppression parameter $\lambda_s$, and the corresponding QGSM results reasonably describe the experimental data for a large energy range, when a relatively small number of incident nucleons participate in the collision (nucleon-nucleus collisions, or peripheral nucleus-nucleus collisions), by thoroughly using the value $\lambda_s$=0.32 to fit those experimental values. The ratio of yields of different particles should not depend, in principle, on the centrality of the collision [1].

# 2 Comparison of QGSM results with experimental data

In Fig. 1 we show the comparison of the QGSM prediction with the experimental data [3] on the dependence of the ratios $\overline{\Omega}^+/\overline{\Lambda}$ (left panel) and $\overline{\Xi}^+$ to $\overline{\Lambda}$ (right panel), on the number of wounded nucleons, $N_\omega$, in Pb+Pb collisions at 158 GeV/c per nucleon. Small values of $N_w$ correspond to peripheral collisions (large impact parameter), while large values of $N_w$ corresponds to central collisions (small impact parameter).

At small values of $N_w$ the ratio is practically equal to that in the cases of p+Be and p+Pb collisions, and they all can be correctly described by the QGSM by using a value of the strangeness

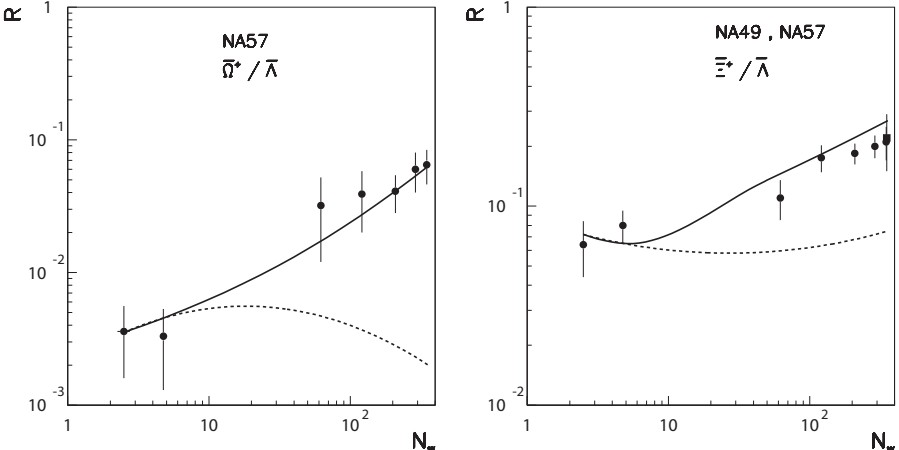

Figure 1: Experimental SPS ratios $\overline{\Omega}^+/\overline{\Lambda}$ (left panel), and of $\overline{\Xi}^+/\overline{\Lambda}$ (right panel), in Pb+Pb collisions, as function of the number of wounded nucleons, $N_w$, compared with the corresponding QGSM results. The full lines show the result of QGSM calculation obtained with the value $\lambda_s$=0.32 at the left end, and with larger values of $\lambda_s$ at its right end, while the dashed lines show the result of QGSM calculation obtained with a constant value $\lambda_s$=0.32, disregarding of the value of the number of wounded nucleons (centrality).

suppression parameter, $\lambda_s$=0.32. However, the experimental ratio increases rather fast with the increasing value of $N_w$, i.e. when we move from peripheral to central Pb+Pb collisions, being central heavy-nuclei collisions the first case in which a different (larger than $\lambda_s$=0.32) value of the strangeness suppression parameter is needed to fit the result of the QGSM calculation to the experimental data on the ratios $R(\overline{\Xi}^+/\overline{\Lambda})$ and $R(\overline{\Omega}^+/\overline{\Lambda})$. This unexpected behaviour for the case of central heavy-nuclei collisions is shown by the full line in Fig. 1, that it has been calculated with the value $\lambda_s$=0.32 at its left end, and with larger values of $\lambda_s$ at its right end (see ref. [1] for details). The results of QGSM calculations with a constant value $\lambda_s$=0.32, disregarding of the value of the number of wounded nucleons (centrality), $N_\omega$, are shown in Fig. 1 by dashed lines.

This behaviour in which the value of the strangeness suppression factor $\lambda_s$ increases with the value of $N_w$ (centrality), indicates (see [1] for details) that the simple quark combinatorial rules are not valid for central collisions of heavy nuclei.

Now we consider the experimental data on midrapidity densities of hyperons in Au+Au collisions measured by the STAR Collaboration [4–6] at RHIC energies.

In Fig. 2 we present the comparison of the QGSM predictions with experimental data on the $N_\omega$ dependence of the ratios $\overline{\Omega}^+/\overline{\Lambda}$ (left panel), and $\overline{\Xi}^+/\overline{\Lambda}$ (right panel), measured by the STAR Collaboration in the midrapidity region, at $\sqrt{s}$ = 62.4 GeV. Similarly as in Fig. 1, the left end of the full line here was calculated with a value $\lambda_s$=0.32, while for the right end larger values of $\lambda_s$ were used [1]. The dashed line was calculated with a constant value of $\lambda_s$=0.32. We see here that the value of $\lambda_s$ that correctly describes multistrange hyperon production at RHIC energies is larger than the one for the case of $\Lambda$ and $\overline{\Lambda}$ production, though this difference between those two values of $\lambda_s$ is not so large as it is for collisions at lower energies. This seems to indicate that the difference in the values of the parameter $\lambda_s$, for multistrange hyperon, and for $\Lambda$ and $\overline{\Lambda}$ production, decreases with the growing of the initial energy of the collision.

In Table 1 we consider the experimental data on $\Xi^-$, $\overline{\Xi}^+$, $\Omega^-$, $\overline{\Omega}^+$ production in central Pb+Pb

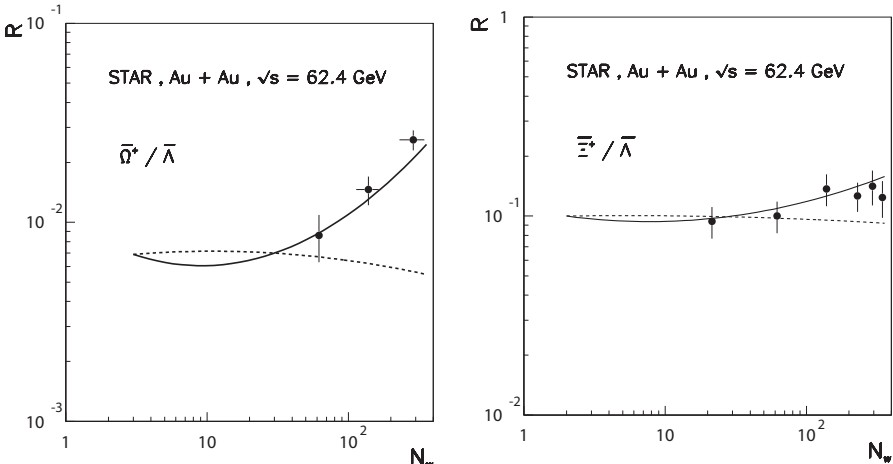

Figure 2: Experimental STAR Collaboration ratios $\overline{\Omega}^+/\overline{\Lambda}$ (left panel), and $\overline{\Xi}^+$ to $\overline{\Lambda}$ (right panel), in Au+Au collisions at $\sqrt{s} = 62.4$ GeV, as function of the number of wounded nucleons, $N_w$, compared with the corresponding QGSM results. The full lines show the result of QGSM calculation obtained with the value $\lambda_s$=0.32 at the left end, and with larger values of $\lambda_s$ at its right end, while the dashed lines show the result of QGSM calculation obtained with a constant value $\lambda_s$=0.32, disregarding of the value of the number of wounded nucleons (centrality).

Table 1: Experimental data on dn/dy of $\Xi^-$, and of $\Xi^-$, $\overline{\Xi}^+$, $\Omega^-$, and $\overline{\Omega}^+$ production in central Pb+Pb collisions at $\sqrt{s}$=2.76 TeV per nucleon by the ALICE Collaboration [7], compared with the corresponding QGSM results.

| Process | $\sqrt{s}$ (TeV) | Centrality | dn/dy (Exp. Data) | dn/dy (QGSM) | $\lambda_s$ |
|---|---|---|---|---|---|
| Pb+Pb $\to \Xi^-$ | 2.76 | 0−10% | $3.34 \pm 0.06 \pm 0.24$ | 3.357 | 0.32 |
| Pb+Pb $\to \overline{\Xi}^+$ | 2.76 | 0−10% | $3.28 \pm 0.06 \pm 0.23$ | 3.317 | |
| Pb+Pb $\to \Omega^-$ | 2.76 | 0−10% | $0.58 \pm 0.04 \pm 0.09$ | 0.606 | 0.38 |
| Pb+Pb $\to \overline{\Omega}^+$ | 2.76 | 0−10% | $0.60 \pm 0.05 \pm 0.09$ | 0.601 | |

collisions at $\sqrt{s}$=2.76 Tev measured by the ALICE Collaboration [7], at the CERN LHC. Now, the strangeness suppression parameter $\lambda_s$ for $\Xi^-$ and $\overline{\Xi}^+$ production becomes smaller than at RHIC energies, taking the standard value $\lambda_s$=0.32. In the case of $\Omega^-$ and $\overline{\Omega}^+$ production, the value of $\lambda_s$ also decreases with respect to the RHIC energy range. Thus we see that the unusually large values of $\lambda_s$ for central Pb+Pb collisions at 158 GeV/c per nucleon, monotonically decrease with the increase of the initial energy of the collision.

## 3 Conclusion

It is shown that the experimental data on the production of hyperons in central collisions of heavy nuclei present a very significant centrality dependence at CERN-SPS energies $\sqrt{s} = 17.3$ GeV. This dependence decreases with the growth of the initial energy of the collision (at RHIC and LHC energies). As for today, there is no any consistent theoretical explanation of this, in principle unexpected, experimental fact, but it must be surely connected to some intrinsic dynamical dif-

ference between the central and the peripheral interactions. Our aim is now to investigate more deeply the dynamics behind this specific behaviour of the central heavy-nuclei collisions.

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
