# Peer review of "Centrality dependence of multistrange baryon production in high-energy heavy-ion collisions"

_SciPost Physics Proceedings_

## Round 1 · Referee Report · Anonymous (Referee 1) · 2022-12-19

Report
The manuscript entitled 'Centrality dependence of multistrange baryon production in high-energy heavy-ion collisions' observes that the strangeness suppression factor in the Quark-Gluon String Model shows an unexpected centrality dependence at SPS and RHIC energies. The manuscript is generally suitable for publication as proceedings to ISMD 2022. I only have a couple of minor comments: * it would be nice to explain the solid and dashed lines in the figure captions * if the authors could briefly explain how the values of lambda_s entering the solid lines are found, that would add to the clarity of the manuscript *it would be very nice if the authors could comment on the physical interpretation of their findings (even if it currently is speculative) * Towards the bottom of page 3: I don't understand the statement starting with 'We see here that the value of lambda_s for multistrange hyperon production...'[until the end of the paragraph] * typos: - last line on page 2: 'and all they can' -> 'and they all can' - first line on page 3: 'incresing' -> 'increasing' - Table 1: some entries have commas instead of points

Author: Carlos Merino on 2023-01-20 [id 3253]
(in reply to Report 1 on 2022-12-19)We have revisited our contribution "Centrality dependence of multistrange baryon production in high-energy heavy-ion collisions" (see adjoint .pdf file), to the Proceedings of the 51st International Symposium on Multiparticle Dynamics (ISMD2022), held in Pitlochry, Scottish Highlands, on 1-5 August 2022, to comply with the minor comments raised by the referee.
Thus is the list of changes we have done in our manuscript to comply with the minor comments raised by the referee:
We have added description of what solid and dashed lines are in the figure captions of figures 1 and 2.
We have added some sentences in the last paragraph of section 1, and in the second paragraph of section 2, to explain how the values of parameter lambda_s entering solid lines are determined.
We have included in the section Conclusion some discussion on where to look for the possible theoretical interpretation of our findings.
At the end of the before-the-last paragraph of section 2, to have rewritten the sentence starting with "We see here (...), and the all the sentences until the end of the paragraph to make the meaning of our statement.
We have corrected all the typos signaled by the referee in her/his report.
All the best,
Carlos Merino (on behalf of the authors)
Carlos Merino Gayoso Dpto. Física de Partículas - Facultad de Física Instituto Galego de Física de Altas Enerxías (IGFAE) Campus Universitario s/n Universidad de Santiago de Compostela 15782 Santiago de Compostela Spain
Attachment:
Contribution_ISMD2022_CarlosMerino_Revisited.pdf

---

## Editorial Decision

unknown